# Mapping Estimation for Discrete Optimal Transport

**Michaël Perrot**
Univ Lyon, UJM-Saint-Etienne, CNRS,
Lab. Hubert Curien UMR 5516, F-42023
michael.perrot@univ-st-etienne.fr

**Nicolas Courty**
Université de Bretagne Sud,
IRISA, UMR 6074, CNRS,
courty@univ-ubs.fr

**Rémi Flamary**
Université Côte d'Azur,
Lagrange, UMR 7293 , CNRS, OCA
remi.flamary@unice.fr

**Amaury Habrard**
Univ Lyon, UJM-Saint-Etienne, CNRS,
Lab. Hubert Curien UMR 5516, F-42023
amaury.habrard@univ-st-etienne.fr

## Abstract

We are interested in the computation of the transport map of an Optimal Transport problem. Most of the computational approaches of Optimal Transport use the Kantorovich relaxation of the problem to learn a probabilistic coupling $\gamma$ but do not address the problem of learning the underlying transport map $T$ linked to the original Monge problem. Consequently, it lowers the potential usage of such methods in contexts where out-of-samples computations are mandatory. In this paper we propose a new way to jointly learn the coupling and an approximation of the transport map. We use a jointly convex formulation which can be efficiently optimized. Additionally, jointly learning the coupling and the transport map allows to smooth the result of the Optimal Transport and generalize it to out-of-samples examples. Empirically, we show the interest and the relevance of our method in two tasks: domain adaptation and image editing.

## 1 Introduction

In recent years Optimal Transport (OT) [1] has received a lot of attention in the machine learning community [2, 3, 4, 5]. This gain of interest comes from several nice properties of OT when used as a divergence to compare discrete distributions: *(i)* it provides a sound and theoretically grounded way of comparing multivariate probability distributions without the need for estimating parametric versions and *(ii)* by considering the geometry of the underlying space through a cost metric, it can encode useful information about the nature of the problem.

OT is usually expressed as an optimal cost functional but it also enjoys a dual variational formulation [1, Chapter 5]. It has been proven useful in several settings. As a first example it corresponds to the Wasserstein distance in the space of probability distributions. Using this distance it is possible to compute means and barycentres [6, 7] or to perform a PCA in the space of probability measures [8]. This distance has also been used in subspace identification problems for analysing the differences between distributions [9], in graph based semi-supervised learning to propagate histogram labels across nodes [4] or as a way to define a loss function for multi-label learning [5]. As a second example OT enjoys a variety of bounds for the convergence rate of empirical to population measures which can be used to derive new probabilistic bounds for the performance of unsupervised learning algorithms such as $k$-means [2]. As a last example OT is a mean of interpolation between distributions [10] that has been used in Bayesian inference [11], color transfer [12] or domain adaptation [13].

On the computational side, despite some results with finite difference schemes [14], one of the major gain is the recent development of regularized versions that leads to efficient algorithms [3, 7, 15]. Most

OT formulations are based on the computation of a (probabilistic) coupling matrix that can be seen as a bi-partite graph between the bins of the distributions. This coupling, also called *transportation matrix*, corresponds to an empirical transport map which suffers from some drawbacks: it can only be applied to the examples used to learn it. In other words when a new dataset (or sample) is available, one has to recompute an OT problem to deal with the new instances which can be prohibitive for some applications in particular when the task is similar or related. From a machine learning standpoint, this also means that we do not know how to find a good approximation of a transport map computed from a small sample that can be generalized to unseen data. This is particularly critical when one considers medium or large scale applications such as image editing problems. In this paper, we propose to bridge this gap by learning an explicit transformation that can be interpreted as a good approximation of the transport map. As far as we know, this is the first approach that addresses directly this problem of *out-of-sample* mapping.

Our formulation is based on classic regularized regression and admits two appealing interpretations. On the one hand, it can be seen as learning a transformation regularized by a transport map. On the other hand, we can see it as the computation of the transport map regularized *w.r.t.* the definition of a transformation (e.g. linear, non-linear, …). This results in an optimization problem that jointly learns both the transport map and the transformation. This formulation can be efficiently solved thanks to alternating block-coordinate descent and actually benefits the two models: *(i)* we obtain smoother transport maps that must be compliant with a transformation that can be used on *out-of-sample* examples and *(ii)* the transformation is able to take into account some geometrical information captured by OT. See Figure 1 for an illustration. We provide some empirical evidence for the usefulness of our approach in domain adaptation and image editing. Beyond that, we think that this paper can open the door to new research on the generalization ability of OT.

The rest of the paper is organized as follows. Section 2 introduces some notations and preliminaries in optimal transport. We present our approach in Section 3. Our experimental evaluation is given in Section 4 and we conclude in Section 5.

## 2   Background

**Monge problem**   Let $\Omega_{\mathcal{S}} \in \mathbb{R}^{d_s}$ and $\Omega_{\mathcal{T}} \in \mathbb{R}^{d_t}$ be two separable metric spaces such that any probability measure on $\Omega_{\mathcal{S}}$, respectively $\Omega_{\mathcal{T}}$, is a Radon measure. By considering a cost function $c : \Omega_{\mathcal{S}} \times \Omega_{\mathcal{T}} \to [0, \infty[$, Monge's formulation of the OT problem is to find a transport map $T : \Omega_{\mathcal{S}} \to \Omega_{\mathcal{T}}$ (also known as a *push-forward* operator) between two probability measures $\mu_{\mathcal{S}}$ on $\Omega_{\mathcal{S}}$ and $\mu_{\mathcal{T}}$ on $\Omega_{\mathcal{T}}$ realizing the infimum of the following function:

$$\inf \left\{ \int_{\Omega_{\mathcal{S}}} c(\mathbf{x}, T(\mathbf{x})) d\mu_{\mathcal{S}}(\mathbf{x}), \ \ T\#\mu_{\mathcal{S}} = \mu_{\mathcal{T}} \right\}. \tag{1}$$

When reaching this infimum, the corresponding map $T$ is an optimal transport map. It associates one point from $\Omega_{\mathcal{S}}$ to a single point in $\Omega_{\mathcal{T}}$. Therefore, the existence of this map is not always guaranteed, as when for example $\mu_{\mathcal{S}}$ is a Dirac and $\mu_{\mathcal{T}}$ is not. As such, the existence of solutions for this problem can in general not be established when $\mu_{\mathcal{S}}$ and $\mu_{\mathcal{T}}$ are supported on a different number of Diracs. Yet, in a machine learning context, data samples usually form discrete distributions, but can be seen as observations of a regular, continuous (with respect to the Lebesgue measure) underlying distribution, thus fulfilling existence conditions (see [1, Chapter 9]). As such, assuming the existence of $T$ calls for a relaxation of the previous problem.

**Kantorovich relaxation**   The Kantorovitch formulation of OT [16] is a convex relaxation of the Monge problem. Let us define $\Pi$ as the set of all probabilistic couplings in $\mathcal{P}(\Omega_{\mathcal{S}} \times \Omega_{\mathcal{T}})$, the space of all joint distributions with marginals $\mu_{\mathcal{S}}$ and $\mu_{\mathcal{T}}$. The Kantorovitch problem seeks for a general coupling $\gamma \in \Pi$ between $\Omega_{\mathcal{S}}$ and $\Omega_{\mathcal{T}}$:

$$\gamma_0 = \underset{\gamma \in \Pi}{\arg\min} \int_{\Omega_{\mathcal{S}} \times \Omega_{\mathcal{T}}} c(\mathbf{x}^s, \mathbf{x}^t) d\gamma(\mathbf{x}^s, \mathbf{x}^t). \tag{2}$$

The optimal coupling always exists [1, Theorem 4.1]. This leads to a simple formulation of the OT problem in the discrete case, *i.e.* whenever $\mu_{\mathcal{S}}$ and $\mu_{\mathcal{T}}$ are only accessible through discrete samples $\mathbf{X}_s = \{\mathbf{x}_i^s\}_{i=1}^{n_s}$, and $\mathbf{X}_t = \{\mathbf{x}_i^t\}_{i=1}^{n_t}$. The corresponding empirical distributions can be written as $\hat{\mu}_{\mathcal{S}} = \sum_{i=1}^{n_s} p_i^s \delta_{\mathbf{x}_i^s}$ and $\hat{\mu}_{\mathcal{T}} = \sum_{i=1}^{n_t} p_i^t \delta_{\mathbf{x}_i^t}$ where $\delta_{\mathbf{x}}$ is the Dirac function at location

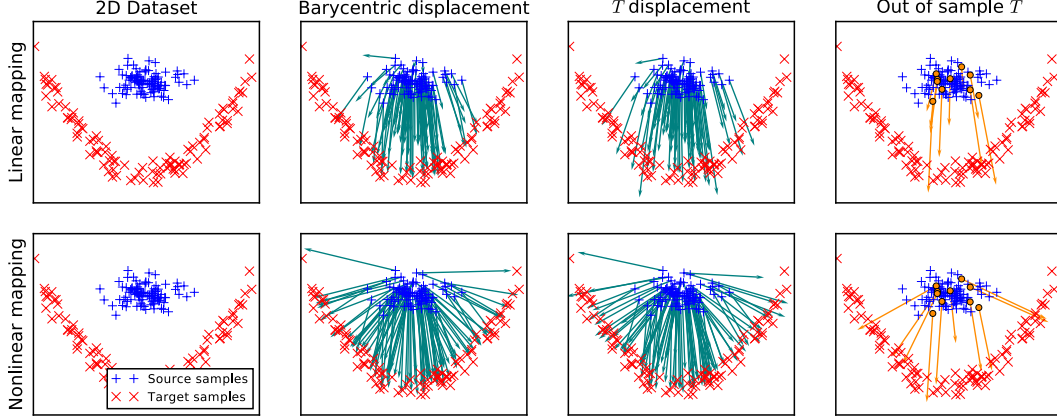

Figure 1: Illustration of the mappings estimated on the clown dataset with a linear (top) and nonlinear (bottom) mapping (best viewed in color).

$\mathbf{x} \in \Omega$. $p_i^s$ and $p_i^t$ are probability masses associated to the $i$-th sample and belong to the probability simplex, *i.e.* $\sum_{i=1}^{n_s} p_i^s = \sum_{i=1}^{n_t} p_i^t = 1$. Let $\hat{\Pi}$ be the set of probabilistic couplings between the two empirical distributions defined as $\hat{\Pi} = \left\{ \gamma \in (\mathbb{R}^+)^{n_s \times n_t} \mid \gamma \mathbf{1}_{n_t} = \hat{\mu}_{\mathcal{S}}, \gamma^T \mathbf{1}_{n_s} = \hat{\mu}_{\mathcal{T}} \right\}$ where $\mathbf{1}_n$ is a $n$-dimensional vector of ones. Problem 2 becomes:

$$\gamma_0 = \underset{\gamma \in \hat{\Pi}}{\arg\min} \quad \langle \gamma, \mathbf{C} \rangle_{\mathcal{F}}, \tag{3}$$

where $\langle \cdot, \cdot \rangle_{\mathcal{F}}$ is the Frobenius dot product[1] and $\mathbf{C} \geq 0$ is the cost matrix related to the function $c$.

**Barycentric mapping** Once the probabilistic coupling $\gamma_0$ has been computed, one needs to map the examples from $\Omega_{\mathcal{S}}$ to $\Omega_{\mathcal{T}}$. This mapping can be conveniently expressed with respect to the set of examples $\mathbf{X}_t$ as the following *barycentric mapping* [11, 13, 12]:

$$\widehat{\mathbf{x}_i^s} = \underset{\mathbf{x} \in \Omega_{\mathcal{T}}}{\arg\min} \quad \sum_{j=1}^{n_t} \gamma_0(i, j) c(\mathbf{x}, \mathbf{x}_j^t), \tag{4}$$

where $\mathbf{x}_i^s$ is a given source sample and $\widehat{\mathbf{x}_i^s}$ is its corresponding image. When the cost function is the squared $\ell_2$ distance, *i.e.* $c(\mathbf{x}, \mathbf{x}') = \|\mathbf{x} - \mathbf{x}'\|_2^2$, this barycentre corresponds to a weighted average and the sample is mapped into the convex hull of the target examples. For all source samples, this barycentric mapping can therefore be expressed as:

$$\widehat{\mathbf{X}_s} = \mathbf{B}_{\gamma_0}(\mathbf{X}_s) = \text{diag}(\gamma_0 \mathbf{1}_{n_t})^{-1} \gamma_0 \mathbf{X}_t. \tag{5}$$

In the rest of the paper we will focus on a uniform sampling, *i.e.* the examples are drawn i.i.d. from $\mu_{\mathcal{S}}$ and $\mu_{\mathcal{T}}$, whence $\widehat{\mathbf{X}_s} = n_s \gamma_0 \mathbf{X}_t$. The main drawback of the mapping (5) is that it does not allow the projection of out-of-sample examples which do not have been seen during the learning process of $\gamma_0$. It means that to transport a new example $\mathbf{x}^s \sim \Omega_{\mathcal{S}}$ one has to compute the coupling matrix $\gamma_0$ again using this new example. Also, while some authors consider specific regularization of $\gamma$ [3, 13] to control the nature of the coupling, inducing specific properties of the transformation $T$ (*i.e.* regularity, divergence free, etc.) is hard to achieve.

In the next section we present a relaxation of the OT problem, which consists in jointly learning $\gamma$ and $T$. We derive the corresponding optimization problem, and show its usefulness in specific scenarios.

## 3 Contributions

### 3.1 Joint learning of $T$ and $\gamma$

In this paper we propose to solve the problem of optimal transport by jointly learning the matrix $\gamma$ and the transformation function $T$. First of all, we denote $\mathcal{H}$ the space of transformations from $\Omega_{\mathcal{T}}$

to $\Omega_{\mathcal{T}}$ and using a slight abuse of notations $\mathbf{X}_s$ and $\mathbf{X}_t$ are matrices where each line is an example respectively drawn from $\Omega_{\mathcal{S}}$ and $\Omega_{\mathcal{T}}$. We propose the following optimisation problem:

$$\underset{T \in \mathcal{H}, \gamma \in \hat{\Pi}}{\arg\min} f(\gamma, T) = \frac{1}{n_s d_t} \|T(\mathbf{X}_s) - n_s \gamma \mathbf{X}_t\|_{\mathcal{F}}^2 + \frac{\lambda_\gamma}{\max(\mathbf{C})} \langle \gamma, \mathbf{C} \rangle_{\mathcal{F}} + \frac{\lambda_T}{d_s d_t} R(T) \qquad (6)$$

where $T(\mathbf{X}_s)$ is a short-hand for the application of $T$ on each example in $\mathbf{X}_s$, $R(\cdot)$ is a regularization term on $T$ and $\lambda_\gamma, \lambda_T$ are hyper-parameters controlling the trade-off between the three terms in the optimization problem. The first term in (6) depends on both $T$ and $\gamma$ and controls the closeness between the transformation induced by $T$ and the barycentric interpolation obtained from $\gamma$. The second term only depends on $\gamma$ and corresponds to the standard optimal transport loss. The third term regularizes $T$ to ensure a better generalization.

A standard approach to solve problem (6) is to use block-coordinate descent (BCD) [17], where the idea is to alternatively optimize for $T$ and $\gamma$. In the next theorem we show that under some mild assumptions on the regularization term $R(\cdot)$ and the function space $\mathcal{H}$ this problem is jointly convex. Note that in this case we are guaranteed to converge to the optimal solution only if we are strictly convex *w.r.t.* $T$ and $\gamma$. While this is not the case for $\gamma$, the algorithm works well in practice and a small regularization term can be added if theoretical convergence is required. The proof of the following theorem can be found in the supplementary.

**Theorem 1.** *Let $\mathcal{H}$ be a convex space and $R(\cdot)$ be a convex function. Problem* (6) *is jointly convex in $T$ and $\gamma$.*

As discussed above we propose to solve optimization problem (6) using a block coordinate descent approach. As such we need to find an efficient way to solve: *(i)* for $\gamma$ when $T$ is fixed and *(ii)* for $T$ when $\gamma$ is fixed. To solve the problem *w.r.t.* $\gamma$ with a fixed $T$, a common approach is to use the Frank-Wolfe algorithm [12, 18]. It is a procedure for solving any convex constrained optimization problem with a convex and continuously differentiable objective function over a compact convex subset of any vector space. This algorithm can find an $\epsilon$ approximation of the optimal solution in $O(1/\epsilon)$ iterations [19]. A detailed algorithm is given in the supplementary material. In the next section we discuss the solution of the minimization *w.r.t.* $T$ with fixed $\gamma$ for different functional spaces.

### 3.2 Choosing $\mathcal{H}$

In the previous subsection we presented our method when considering a general set of transformations $\mathcal{H}$. In this section we propose several possibilities for the choice of a convex set $\mathcal{H}$. On the one hand, we propose to define $\mathcal{H}$ as a set of linear transformations from $\Omega_{\mathcal{S}}$ to $\Omega_{\mathcal{T}}$. On the other hand, using the kernel trick, we propose to consider non-linear transformations. A summary of the approach can be found in Algorithm 1.

**Linear transformations**  A first way to define $\mathcal{H}$ is to consider linear transformations induced by a $d_s \times d_t$ real matrix $\mathbf{L}$:

$$\mathcal{H} = \left\{ T : \exists\, \mathbf{L} \in \mathbb{R}^{d_s \times d_t}, \forall \mathbf{x}^s \in \Omega_{\mathcal{S}}, T(\mathbf{x}^s) = \mathbf{x}^{sT} \mathbf{L} \right\}. \qquad (7)$$

Furthermore, we define $R(T) = \|\mathbf{L} - \mathbf{I}\|_{\mathcal{F}}^2$ where $\mathbf{I}$ is the identity matrix. We choose to bias $\mathbf{L}$ toward $\mathbf{I}$ in order to ensure that the examples are not moved too far away from their initial position. In this case we can rewrite optimization problem (6) as:

$$\underset{\mathbf{L} \in \mathbb{R}^{d_s \times d_t}, \gamma \in \hat{\Pi}}{\arg\min} \frac{1}{n_s d_t} \|\mathbf{X}_s \mathbf{L} - n_s \gamma \mathbf{X}_t\|_{\mathcal{F}}^2 + \frac{\lambda_\gamma}{\max(\mathbf{C})} \langle \gamma, \mathbf{C} \rangle_{\mathcal{F}} + \frac{\lambda_T}{d_s d_t} \|\mathbf{L} - \mathbf{I}\|_{\mathcal{F}}^2. \qquad (8)$$

According to Algorithm 1 a part of our procedure requires to solve optimization problem (8) when $\gamma$ is fixed. One solution is to use the following closed form for $\mathbf{L}$:

$$\mathbf{L} = \left( \frac{1}{n_s d_t} \mathbf{X}_s^T \mathbf{X}_s + \frac{\lambda_T}{d_s d_t} \mathbf{I} \right)^{-1} \left( \frac{1}{n_s d_t} \mathbf{X}_s^T n_s \gamma \mathbf{X}_t + \frac{\lambda_T}{d_s d_t} \mathbf{I} \right) \qquad (9)$$

where $(\cdot)^{-1}$ is the matrix inverse (Moore-Penrose pseudo-inverse when the matrix is singular). In the previous definition of $\mathcal{H}$, we considered non biased linear transformations. However it is sometimes desirable to add a bias to the transformation. The equations being very similar in spirit to the non biased case we refer the interested reader to the supplementary material.

**Algorithm 1:** Joint Learning of $\mathbf{L}$ and $\gamma$.

---

**input** : $\mathbf{X}_s, \mathbf{X}_t$ source and target examples and $\lambda_\gamma, \lambda_T$ hyper parameters.
**output**: $\mathbf{L}, \gamma$.

**1 begin**
**2**    Initialize $k = 0$, $\gamma^0 \in \hat{\Pi}$ and $\mathbf{L}^0 = \mathbf{I}$
**3**    **repeat**
**4**      Learn $\gamma^{k+1}$ solving problem (6) with fixed $\mathbf{L}^k$ using a Frank-Wolfe approach.
**5**      Learn $\mathbf{L}^{k+1}$ using Equation (9), (12) or their biased counterparts with fixed $\gamma^{k+1}$.
**6**      Set $k = k + 1$.
**7**    **until** *convergence*

---

**Non-linear transformations**    In some cases a linear transformation is not sufficient to approximate the transport map. Hence, we propose to consider non-linear transformations. Let $\phi$ be a non-linear function associated to a kernel function $k : \Omega_\mathcal{S} \times \Omega_\mathcal{S} \to \mathbb{R}$ such that $k(\mathbf{x}^s, \mathbf{x}^{s\prime}) = \langle \phi(\mathbf{x}^s), \phi(\mathbf{x}^{s\prime}) \rangle_\mathcal{H}$, we can define $\mathcal{H}$ for a given set of examples $\mathbf{X}_s$ as:

$$\mathcal{H} = \left\{ T : \exists\, \mathbf{L} \in \mathbb{R}^{n^s \times d^t} \forall \mathbf{x}^s \in \Omega_\mathcal{S}, T(\mathbf{x}^s) = k_{\mathbf{X}_s}(\mathbf{x}^{sT})\mathbf{L} \right\} \tag{10}$$

where $k_{\mathbf{X}_s}(\mathbf{x}^{sT})$ is a short-hand for the vector $\left( k(\mathbf{x}^s, \mathbf{x}_1^s) \quad k(\mathbf{x}^s, \mathbf{x}_2^s) \quad \cdots \quad k(\mathbf{x}^s, \mathbf{x}_{n_s}^s) \right)$ where $\mathbf{x}_1^s, \cdots, \mathbf{x}_{n_s}^s \in \mathbf{X}_s$. In this case optimization problem (6) becomes:

$$\underset{\mathbf{L} \in \mathbb{R}^{n^s \times d^t}, \gamma \in \hat{\Pi}}{\arg\min} \frac{1}{n_s d_t} \| k_{\mathbf{X}_s}(\mathbf{X}_s)\mathbf{L} - n_s \gamma \mathbf{X}_t \|_\mathcal{F}^2 + \frac{\lambda_\gamma}{\max(\mathbf{C})} \langle \gamma, \mathbf{C} \rangle_\mathcal{F} + \frac{\lambda_T}{n_s d_t} \| k_{\mathbf{X}_s}(\cdot)\mathbf{L} \|_\mathcal{F}^2. \tag{11}$$

where $k_{\mathbf{X}_s}(\cdot)$ is a short-hand for the vector $\left( k(\cdot, \mathbf{x}_1^s) \quad \cdots \quad k(\cdot, \mathbf{x}_{n_s}^s) \right) = \left( \phi(\mathbf{x}_1^s) \quad \cdots \quad \phi(\mathbf{x}_{n_s}^s) \right)$. As in the linear case there is a closed form solution for $\mathbf{L}$ when $\gamma$ is fixed:

$$\mathbf{L} = \left( \frac{1}{n_s d_t} k_{\mathbf{X}_s}(\mathbf{X}_s) + \frac{\lambda_T}{d^2}\mathbf{I} \right)^{-1} \frac{1}{n_s d_t} n_s \gamma \mathbf{X}_t. \tag{12}$$

As in the linear case it might be interesting to use a bias (Presented in the supplementary material).

### 3.3   Discussion on the quality of the transport map approximation

In this section we propose to discuss some theoretical considerations about our framework and more precisely on the quality of the learned transformation $T$. To assess this quality we consider the Frobenius norm between $T$ and the true transport map, denoted $T^*$, that we would obtain if we could solve Monge's problem. Let $\mathbf{B}_{\hat{\gamma}}$ be the empirical barycentric mapping of $\mathbf{X}_s$ using the probabilistic coupling $\hat{\gamma}$ learned between $\mathbf{X}_s$ and $\mathbf{X}_t$. Similarly let $\mathbf{B}_{\gamma_0}$ be the theoretical barycentric mapping associated with the probabilistic coupling $\gamma_0$ learned on $\mu_\mathcal{S}, \mu_\mathcal{T}$ the whole distributions and which corresponds to the solution of Kantorovich's problem. Using a slight abuse of notations we denote by $\mathbf{B}_{\hat{\gamma}}(\mathbf{x}^s)$ and $\mathbf{B}_{\gamma_0}(\mathbf{x}^s)$ the projection of $\mathbf{x}^s \in \mathbf{X}_s$ by these barycentric mappings. Using the triangle inequality, some standard properties on the square function, the definition of $\mathcal{H}$ and [20, Theorem 2], we have with high probability that (See the supplementary material for a justification):

$$\underset{\mathbf{x}^s \sim \Omega_\mathcal{S}}{\mathbb{E}} \| T(\mathbf{x}^s) - T^*(\mathbf{x}^s) \|_\mathcal{F}^2 \le 4 \sum_{\mathbf{x}^s \in \mathbf{X}_s} \| T(\mathbf{x}^s) - \mathbf{B}_{\hat{\gamma}}(\mathbf{x}^s) \|_\mathcal{F}^2 + \mathcal{O}\left( \frac{1}{\sqrt{n_s}} \right)$$

$$+ 4 \sum_{\mathbf{x}^s \in \mathbf{X}_s} \| \mathbf{B}_{\hat{\gamma}}(\mathbf{x}^s) - \mathbf{B}_{\gamma_0}(\mathbf{x}^s) \|_\mathcal{F}^2 + 2 \underset{\mathbf{x}^s \sim \Omega_\mathcal{S}}{\mathbb{E}} \| \mathbf{B}_{\gamma_0}(\mathbf{x}^s) - T^*(\mathbf{x}^s) \|_\mathcal{F}^2. \tag{13}$$

From Inequality 13 we assess the quality of the learned transformation $T$ *w.r.t.* three key quantities. The first quantity, $\sum_{\mathbf{x}^s \in \mathbf{X}_s} \| T(\mathbf{x}^s) - \mathbf{B}_{\hat{\gamma}}(\mathbf{x}^s) \|_\mathcal{F}^2$, is a measure of the difference between the learned transformation and the empirical barycentric mapping. We minimize it in Problem (6). The second and third quantities are theoretical and hard to bound because, as far as we know, there is a lack of theoretical results related to these terms in the literature. Nevertheless, we expect $\sum_{\mathbf{x}^s \in \mathbf{X}_s} \| \mathbf{B}_{\hat{\gamma}}(\mathbf{x}^s) - \mathbf{B}_{\gamma_0}(\mathbf{x}^s) \|_\mathcal{F}^2$ to decrease uniformly with respect to the number of examples as it corresponds to a measure of how well the empirical barycentric mapping estimates the theoretical one. Similarly, we expect $\mathbb{E}_{\mathbf{x}^s \sim \Omega_\mathcal{S}} \| \mathbf{B}_{\gamma_0}(\mathbf{x}^s) - T^*(\mathbf{x}^s) \|_\mathcal{F}^2$ to be small as it characterizes that the theoretical barycentric mapping is a good approximation of the true transport map. This depends of course on the expressiveness of the set $\mathcal{H}$ considered. We think that this discussion opens up new theoretical perspectives for OT in Machine Learning but these are beyond the scope of this paper.

Table 1: Accuracy on the Moons dataset. Color-code: the darker the result, the better.

| Angle | 1NN | GFK | SA | OT | L1L2 | OTE | OTLin | | OTLinB | | OTKer | | OTKerB | |
|---|---|---|---|---|---|---|---|---|---|---|---|---|---|---|
| | | | | | | | $T$ | $\gamma$ | $T$ | $\gamma$ | $T$ | $\gamma$ | $T$ | $\gamma$ |
| 10 | **100.0** | 99.9 | **100.0** | 97.9 | 99.6 | **100.0** | 100.0 | 100.0 | 100.0 | 100.0 | 100.0 | 100.0 | 100.0 | 100.0 |
| 20 | 93.1 | 95.8 | 93.1 | 95.0 | 98.7 | **100.0** | 100.0 | 100.0 | 100.0 | 100.0 | 100.0 | 100.0 | 100.0 | 100.0 |
| 30 | 84.0 | 92.5 | 84.0 | 90.6 | 98.4 | **100.0** | 99.8 | 99.9 | 99.8 | 99.9 | 100.0 | 100.0 | 100.0 | 100.0 |
| 40 | 77.1 | 90.8 | 74.4 | 83.7 | 95.8 | **100.0** | 98.3 | 98.7 | 98.1 | 98.5 | 99.7 | 99.7 | 99.6 | 99.7 |
| 50 | 61.7 | 90.2 | 73.1 | 77.8 | 87.7 | 87.3 | 97.8 | 97.6 | 97.5 | 97.5 | 99.1 | **99.2** | 99.1 | 99.1 |
| 60 | 41.2 | 79.4 | 72.3 | 71.0 | 88.3 | 86.3 | 96.4 | **97.2** | 95.8 | 97.0 | 96.6 | 96.8 | 96.6 | 96.8 |
| 70 | 23.1 | 61.0 | 72.3 | 64.5 | 89.0 | 77.5 | 88.0 | **94.7** | 88.2 | 94.3 | 80.8 | 81.5 | 82.5 | 83.1 |
| 80 | 20.7 | 36.2 | 72.3 | 57.3 | 73.6 | 58.8 | 76.9 | **81.0** | 76.6 | 80.7 | 74.0 | 74.1 | 73.9 | 74.2 |
| 90 | 19.4 | 43.1 | 34.2 | 51.0 | 58.1 | 51.3 | 67.9 | 68.0 | 67.1 | **68.1** | 56.3 | 55.8 | 57.6 | 55.4 |

## 4 Experiments

### 4.1 Domain Adaptation

**Datasets** We consider two domain adaptation (DA) datasets, namely Moons [21] and Office-Caltech [22]. The Moons dataset is a binary classification task where the source domain corresponds to two intertwined moons, each one representing a class. The target domain is built by rotating the source domain with an angle ranging from 10 to 90 degrees. It leads to 9 different adaptation tasks of increasing difficulty. The examples are two dimensional and we consider 300 source and target examples for training and 1000 target examples for testing. The Office-Caltech dataset is a 10 class image classification task with 4 domains corresponding to images coming from different sources: amazom (A), dslr (D), webcam (W) and Caltech10 (C). There are 12 adaptation tasks where each domain is in turn considered as the source or the target (denoted source $\rightarrow$ target). To represent the images we use the deep learning features of size 4096 named decaf6 [23]. During the training process we consider all the examples from the source domain and half of the examples from the target domain, the other half being used as the test set.

**Methods** We consider 6 baselines. The first one is a simple 1-Nearest-Neighbour (1NN) using the original source examples only. The second and third ones are two widely used DA approaches, namely Geodesic Flow Kernel (GFK) [22] and Subspace Alignment (SA) [24]. The fourth to sixth baselines are OT based approaches: the classic OT method (OT), OT with entropy based regularization (OTE) [3] and OT with $\ell_1\ell_2$ regularization (L1L2) [13]. We present the results of our approach with the linear (OTLin) and kernel (OTKer) versions of $T$ and their biased counterpart (*B). For OT based methods the idea is to *(i)* compute the transport map between the source and the target, *(ii)* project the source examples and *(iii)* classify the target examples using a 1NN on the projected source.

**Experimental Setup** We consider the following experimental setup for all the methods and datasets. All the results presented in this section are averaged over 10 trials. For each trial we consider three sets of examples, a labelled source training set denoted $\mathbf{X}_s, \mathbf{y}_s$, an unlabelled target training set denoted $\mathbf{X}_t^{train}$ and a labelled target testing set $\mathbf{X}_t^{test}$. The model is learned on $\mathbf{X}_s, \mathbf{y}_s$ and $\mathbf{X}_t^{train}$ and evaluated on $\mathbf{X}_t^{test}$ with a 1NN learned on $\mathbf{X}_s, \mathbf{y}_s$. All the hyper-parameters are tuned according to a grid search on the source and target training instances using a circular validation procedure derived from [21, 25] and described in the supplementary material. For GFK and SA we choose the dimension of the subspace $d \in \{3, 6, \ldots, 30\}$, for L1L2 and OTE we set the parameter for entropy regularization in $\{10^{-6}, 10^{-5}, \ldots, 10^5\}$, for L1L2 we choose the class related parameter $\eta \in \{10^{-5}, 10^{-4}, \ldots, 10^2\}$, for all our methods we choose $\lambda_T, \lambda_\gamma \in \{10^{-3}, 10^{-2}, \ldots, 10^0\}$.

The results on the Moons and Office-Caltech datasets are respectively given in Table 1 and 2. A first important remark is that the coupling $\gamma$ and the transformation $T$ almost always obtain the same results. It shows that our method is able to learn a good approximation $T$ of the transport map induced by $\gamma$. In terms of accuracy our approach tends to give the best results. It shows that we are effectively able to move closer the distributions in a relevant way. For the Moons dataset, the last 6 approaches (including ours) based on OT obtain similar results until 40 degrees while the other methods fail to obtain good results at 20 degrees. Beyond 50 degrees, our approaches give significantly better results than the others. Furthermore they are more stable when the difficulty of the problem increases which

Table 2: Accuracy on the Office-Caltech dataset. Color-code: the darker the result, the better.

| Task | 1NN | GFK | SA | OT | L1L2 | OTE | OTLin | | OTLinB | | OTKer | | OTKerB | |
|---|---|---|---|---|---|---|---|---|---|---|---|---|---|---|
| | | | | | | | $T$ | $\gamma$ | $T$ | $\gamma$ | $T$ | $\gamma$ | $T$ | $\gamma$ |
| $D \to W$ | 89.5 | 93.3 | 95.6 | 77.0 | 95.7 | 95.7 | 97.3 | 97.3 | 97.3 | 97.3 | 98.4 | **98.5** | **98.5** | 98.5 |
| $D \to A$ | 62.5 | 77.2 | 88.5 | 70.8 | 74.9 | 74.8 | 85.7 | 85.7 | 85.8 | 85.8 | **89.9** | **89.9** | 89.5 | 89.5 |
| $D \to C$ | 51.8 | 69.7 | **79.0** | 68.1 | 67.8 | 68.0 | 77.2 | 77.2 | 77.4 | 77.4 | 69.1 | 69.2 | 69.3 | 69.3 |
| $W \to D$ | 99.2 | **99.8** | 99.6 | 74.1 | 94.4 | 94.4 | 99.4 | 99.4 | **99.8** | **99.8** | 97.2 | 97.2 | 96.9 | 96.9 |
| $W \to A$ | 62.5 | 72.4 | 79.2 | 67.6 | 71.3 | 71.3 | **81.5** | **81.5** | 81.4 | 81.4 | 78.5 | 78.3 | 78.5 | 78.8 |
| $W \to C$ | 59.5 | 63.7 | 55.0 | 63.1 | 67.8 | 67.8 | **75.9** | **75.9** | 75.4 | 75.4 | 72.7 | 72.7 | 65.1 | 63.3 |
| $A \to D$ | 65.2 | 75.9 | **83.8** | 64.6 | 70.1 | 70.5 | 80.6 | 80.6 | 80.4 | 80.5 | 65.6 | 65.5 | 71.9 | 71.5 |
| $A \to W$ | 56.8 | 68.0 | 74.6 | 66.8 | 67.2 | 67.3 | **74.6** | **74.6** | 74.4 | 74.4 | 66.4 | 64.8 | 70.0 | 68.9 |
| $A \to C$ | 70.1 | 75.7 | 79.2 | 70.4 | 74.1 | 74.3 | 81.8 | 81.8 | 81.6 | 81.6 | 84.4 | 84.4 | **84.5** | **84.5** |
| $C \to D$ | 75.9 | 79.5 | 85.0 | 66.0 | 69.8 | 70.2 | 87.1 | 87.1 | **87.2** | **87.2** | 70.1 | 70.0 | 78.6 | 78.6 |
| $C \to W$ | 65.2 | 70.7 | 74.4 | 59.2 | 63.8 | 63.8 | 78.3 | 78.3 | 78.5 | 78.5 | 80.0 | **80.4** | 73.5 | 73.4 |
| $C \to A$ | 85.8 | 87.1 | 89.3 | 75.2 | 76.6 | 76.7 | **89.9** | **89.9** | 89.7 | 89.7 | 82.4 | 82.2 | 83.6 | 83.5 |
| Mean | 70.3 | 77.8 | 81.9 | 68.6 | 74.5 | 74.6 | **84.1** | **84.1** | **84.1** | **84.1** | 79.6 | 79.4 | 80.0 | 79.7 |

can be interpreted as a benefit from our regularization. In the supplementary material we propose an illustration of the transformation learned by our approach. For Office-Caltech, our methods are significantly better than other approaches which illustrates the potential of our method for difficult tasks. To conclude, forcing OT to simultaneously learn coupling and transformation seems beneficial.

## 4.2 Seamless copy in images with gradient adaptation

We propose here a direct application of our mapping estimation in the context of image editing. While several papers using OT are focusing on color adaptation [12, 26], we explore here a new variant in the domain of image editing: the seamless editing or cloning in images. In this context, one may desire to import a region from a given source image to a target image. As a direct copy of the region leads to inaccurate results in the final image nearby the boundaries of the copied selection, a very popular method, proposed by Pérez and co-workers [27], allows to seamlessly blend the target image and the selection. This technique, coined as *Poisson Image Editing*, operates in the gradient domain of the image. Hence, the gradients of the selection operate as a guidance field for an image reconstruction based on membrane interpolation with appropriate boundary conditions extracted from the target image (See the supplementary material for more details).

Though appealing, this technique is prone to errors due local contrast change or false colors resulting from the integration. While some solutions combining both gradient and color domains exist [28], this editing technique usually requires the source and target images to have similar colors and contrast. Here, we propose to enhance the genericity of this technique by forcing the gradient distribution from the source image to follow the gradient distribution in the target image. As a result, the seamless cloning not only blends smoothly the copied region in the target domain, but also constrains the color dynamics to that of the target image. Hence, a part of the style of the target image is preserved. We start by learning a transfer function $T_{s \to t} : \mathbb{R}^6 \to \mathbb{R}^6$ with our method, where 6 denotes the vertical and horizontal components of gradient per color, and we then directly solve the same system as [27].

When dealing with images, the number of source and target gradients are largely exceeding tens of thousands and it is mandatory to consider methods that scale appropriately. As such, our technique can readily learn the transfer function $T_{s \to t}$ over a limited set of gradients and generalizes appropriately to unseen gradients. Three illustrations of this method are proposed in a context of face swapping in Figure 2. As one can observe, the original method of Poisson image editing [27] (3rd column) tends to preserve the color dynamics of the original image and fails in copying the style of the target image. Our method was tested with a linear and kernel version of $T_{s \to t}$, that was learned with only 500 gradients sampled randomly from both sources ($\lambda_T = 10^{-2}$, $\lambda_T = 10^3$ for respectively the linear and kernel versions, and $\lambda_\gamma = 10^{-7}$ for both cases). As a general qualitative comment, one can observe that the kernel version of $T_{s \to t}$ is better at preserving the dynamics of the gradient, while the linear version tends to flatten the colors. In this low-dimensional space, this illustrates the need of a non-linear transformation. Regarding the computational time, the gradient adaptation is of the same

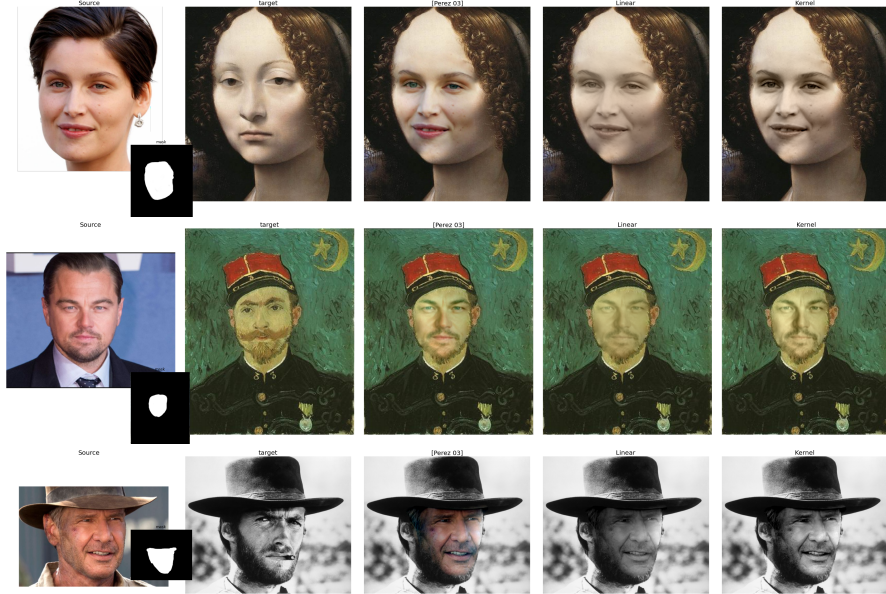

Figure 2: Illustrations of seamless copies with gradient adaptation. Each row is composed of the source image, the corresponding selection zone $\Omega$ described as a binary mask, and the target image. We compare here the two linear (4th column) and kernel (5th column) versions of the map $T_{s \rightarrow t}$ with the original method of [27] (2nd column) (best viewed in color).

order of magnitude as the Poisson equation solving, and each example is computed in less than 30s on a standard personal laptop. In the supplementary material we give other examples of the method.

## 5   Conclusion

In this paper we proposed a jointly convex approach to learn both the coupling $\gamma$ and a transformation $T$ approximating the transport map given by $\gamma$. It allowed us to apply a learned transport to a set of out-of-samples examples not seen during the learning process. Furthermore, jointly learning the coupling and the transformation allowed us to regularize the transport by enforcing a certain smoothness on the transport map. We also proposed several possibilities to choose $\mathcal{H}$ the set of possible transformations. We presented some theoretical considerations on the generalization ability of the learned transformation $T$. Hence we discussed that under the assumption that the barycentric mapping generalizes well and is a good estimate of the true transformation, then $T$ learned with our method should be a good approximation of the true transformation. We have shown that our approach is efficient in practice on two different tasks: domain adaptation and image editing.

The framework presented in this paper opens the door to several perspectives. First, from a theoretical standpoint the bound proposed raises some questions on the generalization ability of the barycentric mapping and on the estimation of the quality of the true barycentric mapping with respect to the target transformation. On a more practical side, note that in recent years regularized OT has encountered a growing interest and several methods have been proposed to control the behaviour of the transport. As long as these regularization terms are convex, one could imagine using them in our framework. Another perspective could be to use our framework in a mini-batch setting where instead of learning from the whole dataset we can estimate a single function $T$ from several couplings $\gamma$ optimized on different splits of the examples. As a last example we believe that our framework could allow the use of the notion of OT in deep architectures as, contrary to the coupling $\gamma$, the function $T$ can be used on out-of-samples examples.

**Acknowledgments**

This work was supported in part by the french ANR project LIVES ANR-15-CE23-0026-03.

## Footnotes

[1] $\langle \mathbf{A}, \mathbf{B} \rangle_{\mathcal{F}} = \text{Tr}(\mathbf{A}^T \mathbf{B})$

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
