[Supplementary Material]

# Mapping Estimation for Discrete Optimal Transport
# Supplementary Material

**Michaël Perrot**
Univ Lyon, UJM-Saint-Etienne, CNRS,
Lab. Hubert Curien UMR 5516, F-42023
`michael.perrot@univ-st-etienne.fr`

**Nicolas Courty**
Université de Bretagne Sud,
IRISA, UMR 6074, CNRS,
`courty@univ-ubs.fr`

**Rémi Flamary**
Université Côte d'Azur,
Lagrange, UMR 7293 , CNRS, OCA
`remi.flamary@unice.fr`

**Amaury Habrard**
Univ Lyon, UJM-Saint-Etienne, CNRS,
Lab. Hubert Curien UMR 5516, F-42023
`amaury.habrard@univ-st-etienne.fr`

## 1 Proof of the Joint Convexity of the Optimization Problem

We recall the optimization problem:

$$\underset{T \in \mathcal{H}, \gamma \in \hat{\Pi}}{\arg\min} f(\gamma, T) = \frac{1}{n_s d_t} \|T(\mathbf{X}_s) - n_s \gamma \mathbf{X}_t\|_{\mathcal{F}}^2 + \frac{\lambda_\gamma}{\max(\mathbf{C})} \langle \gamma, \mathbf{C} \rangle_{\mathcal{F}} + \frac{\lambda_T}{d_s d_t} R(T) \qquad (1)$$

and the theorem:

**Theorem 1.** *Let $\mathcal{H}$ be a convex space and $R(\cdot)$ be a convex function. Problem* (1) *is jointly convex in $T$ and $\gamma$.*

*Proof.* First of all recall that a sum of jointly convex functions is jointly convex. Hence it is sufficient to show that the three terms of optimization problem (1) are jointly convex. We note $f_1(\gamma, T) = \frac{1}{n_s d_t} \|T(\mathbf{X}_s) - n_s \gamma \mathbf{X}_t\|_{\mathcal{F}}^2$, $f_2(\gamma) = \frac{\lambda_\gamma}{\max(\mathbf{C})} \langle \gamma, \mathbf{C} \rangle_{\mathcal{F}}$ and $f_3(T) = \frac{\lambda_T}{d_s d_t} R(T)$. $f_1$ depends on both $T$ and $\gamma$ and controls the proximity between the transformation induced by $T$ and the barycentric interpolation obtained from $\gamma$. $f_2$ only depends on $\gamma$, it corresponds to the standard term minimized to solve the optimal transport problem. $f_3$ regularizes $T$ to ensure a better generalization.

Note that $f_2$ and $f_3$ are by construction jointly convex in $\gamma$ and $T$. We will show that the $f_1$ is also jointly convex. Let $g(\gamma, T) = \|T(\mathbf{X}_s) - n_s \gamma \mathbf{X}_t\|_{\mathcal{F}}$, we want to show that:

$$g(t\gamma_1 + (t-1)\gamma_2, tT_1 + (1-t)T_2) \le tg(\gamma_1, T_1) + (1-t)g(\gamma_2, T_2).$$

We have:

$$\|(tT_1 + (1-t)T_2)(\mathbf{X}_s) - n_s(t\gamma_1 + (t-1)\gamma_2)\mathbf{X}_t\|_{\mathcal{F}}$$

(Triangle inequality and definition of $\mathcal{H}$.)

$$\le \|tT_1(\mathbf{X}_s) - tn_s\gamma_1\mathbf{X}_t\|_{\mathcal{F}} + \|(1-t)T_2(\mathbf{X}_s) - (1-t)n_s\gamma_2\mathbf{X}_t\|_{\mathcal{F}}$$

($t \in [0,1]$.)

$$= t \|T_1(\mathbf{X}_s) - n_s\gamma_1\mathbf{X}_t\|_{\mathcal{F}} + (1-t) \|T_2(\mathbf{X}_s) - n_s\gamma_2\mathbf{X}_t\|_{\mathcal{F}}$$

Furthermore noting that $g$ is convex and positive we have:

$$[g(t\gamma_1 + (t-1)\gamma_2, tT_1 + (1-t)T_2)]^2$$

$$(\forall x \in \mathbb{R}^+, x \to x^2 \text{ is non decreasing.})$$

$$\leq [tg(\gamma_1, T_1) + (1-t)g(\gamma_2, T_2)]^2$$

$$(\forall x \in \mathbb{R}, x \to x^2 \text{ is convex.})$$

$$\leq t[g(\gamma_1, T_1)]^2 + (1-t)[g(\gamma_2, T_2)]^2.$$

Noting that $f_1(\gamma, T) = \frac{1}{n_s d_t} g(\gamma, T)^2$ concludes the proof. $\qquad\square$

## 2 Details about Block Coordinate Descent

To solve optimization problem (1) we propose to use a block-coordinate descent approach. As such we need to find an efficient way to solve for $\gamma$ when $T$ is fixed and to solve for $T$ when $\gamma$ is fixed.

**Solving for $\gamma$ with $T$ fixed** In this case we want to solve:

$$\underset{\gamma \in \hat{\Pi}}{\arg\min} f(\gamma, T) = \frac{1}{n_s d_t} \|T(\mathbf{X}_s) - n_s\gamma\mathbf{X}_t\|_{\mathcal{F}}^2 + \frac{\lambda_\gamma}{\max(\mathbf{C})} \langle \gamma, \mathbf{C} \rangle_{\mathcal{F}} + \frac{\lambda_T}{d_s d_t} R(T) \qquad (2)$$

where $T$ is the current transformation. To solve such an optimization problem a common approach is to use the Frank-Wolfe algorithm [1, 2]. It is a procedure for solving any convex constrained optimization problems with a convex and continuously differentiable objective function over a compact convex subset of any vector space. This algorithm can find an $\epsilon$ approximation of the optimal solution in $O(1/\epsilon)$ iterations [3]. A detailed algorithm is given in Section 3.

**Solving for $T$ with $\gamma$ fixed** In this case we want to solve:

$$\underset{T \in \mathcal{H}}{\arg\min} f(\gamma, T) = \frac{1}{n_s d_t} \|T(\mathbf{X}_s) - n_s\gamma\mathbf{X}_t\|_{\mathcal{F}}^2 + \frac{\lambda_\gamma}{\max(\mathbf{C})} \langle \gamma, \mathbf{C} \rangle_{\mathcal{F}} + \frac{\lambda_T}{d_s d_t} R(T) \qquad (3)$$

where $\gamma$ is the current mapping between the examples. The solution to this optimization problem depends on the form of $\mathcal{H}$ and $R$. This is discussed in detail in Section 3.2 in the main paper.

## 3 Detailed Frank-Wolfe algorithm

We propose in Algorithm 1 a detailed version of the Frank-Wolfe approach for solving problem (2).

---

**Algorithm 1:** Updating $\gamma$ with the Frank-Wolfe algorithm.

> **input** : The current values of $\gamma$ and $T$.
> **output** : The new value of $\gamma$.
> **1 begin**
> **2**      Initialize $k = 0$ and $\gamma^0 = \gamma$
> **3**      **repeat**
> **4**          Solve $\mathbf{S}^k = \arg\min_{\mathbf{S} \in \hat{\Pi}} \langle \mathbf{S}, \nabla f(\gamma^k, T) \rangle_{\mathcal{F}}$ with
>              $\nabla f(\gamma, T) = \frac{\lambda_\gamma}{\max(\mathbf{C})}\mathbf{C} - \frac{2}{n_s d_t}n_s T(\mathbf{X}_s)\mathbf{X}_t^T + \frac{2}{n_s d_t}n_s^2\gamma\mathbf{X}_t\mathbf{X}_t^T$.
> **5**          Find the optimal step $\alpha^k$ satisfying the Armijo rule that minimizes $f\left((1-\alpha)\gamma^k + \alpha\mathbf{S}^k, T\right)$.
> **6**          Update $\gamma^{k+1} = (1-\alpha)\gamma^k + \alpha\mathbf{S}^k$ and $k = k+1$.
> **7**      **until** *convergence*

---

## 4 Bias including version of $\mathcal{H}$

We present the bias including version of $\mathcal{H}$ both in the linear and the non-linear case.

**Biased linear transformations**  In the biased linear case we have:

$$\mathcal{H} = \left\{ T : \exists\, \mathbf{L} \in \mathbb{R}^{d^s \times d^t}, \exists\, \mathbf{b} \in \mathbb{R}^{d^t}, \forall \mathbf{x} \in \Omega_{\mathcal{S}}, T(\mathbf{x}^s) = \mathbf{x}^{s\,T}\mathbf{L} + \mathbf{b}^T = \begin{pmatrix} \mathbf{x}^{s\,T} & 1 \end{pmatrix} \begin{pmatrix} \mathbf{L} \\ \mathbf{b}^T \end{pmatrix} \right\}. \tag{4}$$

In this case, optimization problem 1 becomes:

$$\underset{\begin{pmatrix} \mathbf{L} \\ \mathbf{b}^T \end{pmatrix} \in \mathbb{R}^{d^s+1 \times d^t}, \gamma \in \hat{\Pi}}{\arg\min} \frac{1}{n_s d_t} \left\| \begin{pmatrix} \mathbf{X}_s & \mathbf{1} \end{pmatrix} \begin{pmatrix} \mathbf{L} \\ \mathbf{b}^T \end{pmatrix} - n_s \gamma \mathbf{X}_t \right\|_{\mathcal{F}}^2 + \frac{\lambda_\gamma}{\max(\mathbf{C})} \langle \gamma, \mathbf{C} \rangle_{\mathcal{F}} + \frac{\lambda_T}{d_s d_t} \|\mathbf{L} - \mathbf{I}\|_{\mathcal{F}}^2. \tag{5}$$

As in the non biased case, it is possible to find a closed form solution for $\begin{pmatrix} \mathbf{L} \\ \mathbf{b}^T \end{pmatrix}$ when $\gamma$ is fixed:

$$\begin{pmatrix} \mathbf{L} \\ \mathbf{b}^T \end{pmatrix} = \left( \frac{1}{n_s d_t} \begin{pmatrix} \mathbf{X}_s \\ \mathbf{1}^T \end{pmatrix} \begin{pmatrix} \mathbf{X}_s & \mathbf{1} \end{pmatrix} + \frac{\lambda_T}{d_s d_t} \begin{pmatrix} \mathbf{I} & \mathbf{0} \\ \mathbf{0}^T & 0 \end{pmatrix} \right)^{-1} \left( \frac{1}{n_s d_t} \begin{pmatrix} \mathbf{X}_s \\ \mathbf{1}^T \end{pmatrix} n_s \gamma \mathbf{X}_t + \frac{\lambda_T}{d_s d_t} \begin{pmatrix} \mathbf{I} \\ \mathbf{0}^T \end{pmatrix} \right). \tag{6}$$

**Biased non-linear transformations**  In the biased non-linear case $\mathcal{H}$ becomes:

$$\mathcal{H} = \left\{ T : \exists\, \mathbf{L} \in \mathbb{R}^{n^s \times d^t}, \exists\, \mathbf{b} \in \mathbb{R}^{d^t}, \forall \mathbf{x}^s \in \Omega_{\mathcal{S}}, T(\mathbf{x}^s) = \begin{pmatrix} k_{\mathbf{X}_s}(\mathbf{x}^{s\,T}) & 1 \end{pmatrix} \begin{pmatrix} \mathbf{L} \\ \mathbf{b}^T \end{pmatrix} \right\} \tag{7}$$

Optimization problem 1 can be rewritten as:

$$\underset{\begin{pmatrix} \mathbf{L} \\ \mathbf{b}^T \end{pmatrix} \in \mathbb{R}^{n^s+1 \times d^t}, \gamma \in \hat{\Pi}}{\arg\min} \frac{1}{n_s d_t} \left\| \begin{pmatrix} k_{\mathbf{x}_s}(\mathbf{X}_s) & 1 \end{pmatrix} \begin{pmatrix} \mathbf{L} \\ \mathbf{b}^T \end{pmatrix} - n_s \gamma \mathbf{X}_t \right\|_{\mathcal{F}}^2 + \frac{\lambda_\gamma}{\max(\mathbf{C})} \langle \gamma, \mathbf{C} \rangle_{\mathcal{F}} + \frac{\lambda_T}{d_s d_t} \|k_{\mathbf{x}_s}(\cdot)\mathbf{L}\|_{\mathcal{F}}^2. \tag{8}$$

As in the non biased case, it is possible to find a closed form solution for $\begin{pmatrix} \mathbf{L} \\ \mathbf{b}^T \end{pmatrix}$ when $\gamma$ is fixed:

$$\begin{pmatrix} \mathbf{L} \\ \mathbf{b}^T \end{pmatrix} = \left( \frac{1}{n_s d_t} \begin{pmatrix} \mathbf{K}_{\mathbf{X}_s \mathbf{X}_s} \\ \mathbf{1}^T \end{pmatrix} \begin{pmatrix} \mathbf{K}_{\mathbf{X}_s \mathbf{X}_s} & \mathbf{1} \end{pmatrix} + \frac{\lambda_T}{d_s d_t} \begin{pmatrix} \mathbf{K}_{\mathbf{X}_s \mathbf{X}_s} & \mathbf{0} \\ \mathbf{0}^T & 0 \end{pmatrix} \right)^{-1} \frac{1}{n_s d_t} \begin{pmatrix} \mathbf{K}_{\mathbf{X}_s \mathbf{X}_s} \\ \mathbf{1}^T \end{pmatrix} n_s \gamma \mathbf{X}_t. \tag{9}$$

## 5  Proof of Equation (13) in the Main Paper

We recall the notations. Let $T^*$ be the true transport map that we would obtain if we could solve Monge's problem. Let $\mathbf{B}_{\hat{\gamma}}$ be the empirical barycentric mapping of $\mathbf{X}_s$ using the probabilistic coupling $\hat{\gamma}$ learned between $\mathbf{X}_s$ and $\mathbf{X}_t$. Similarly let $\mathbf{B}_{\gamma_0}$ be the theoretical barycentric mapping associated with the probabilistic coupling $\gamma_0$ learned on $\mu_{\mathcal{S}}, \mu_{\mathcal{T}}$ the whole distributions and which corresponds to the solution of Kantorovich's problem. Using a slight abuse of notations we denote by $\mathbf{B}_{\hat{\gamma}}(\mathbf{x}^s)$ and $\mathbf{B}_{\gamma_0}(\mathbf{x}^s)$ the projection of $\mathbf{x}^s \in \mathbf{X}_s$ by these barycentric mappings. We have that:

$$\underset{\mathbf{x}^s \sim \Omega_{\mathcal{S}}}{\mathbb{E}} \|T(\mathbf{x}^s) - T^*(\mathbf{x}^s)\|_{\mathcal{F}}^2$$

(Triangle inequality.)

$$\leq \underset{\mathbf{x}^s \sim \Omega_{\mathcal{S}}}{\mathbb{E}} \left( \|T(\mathbf{x}^s) - \mathbf{B}_{\gamma_0}(\mathbf{x}^s)\|_{\mathcal{F}} + \|\mathbf{B}_{\gamma_0}(\mathbf{x}^s) - T^*(\mathbf{x}^s)\|_{\mathcal{F}} \right)^2$$

$$((a+b)^2 \leq 2a^2 + 2b^2.)$$

$$\leq 2 \underset{\mathbf{x}^s \sim \Omega_{\mathcal{S}}}{\mathbb{E}} \|T(\mathbf{x}^s) - \mathbf{B}_{\gamma_0}(\mathbf{x}^s)\|_{\mathcal{F}}^2 + 2 \underset{\mathbf{x}^s \sim \Omega_{\mathcal{S}}}{\mathbb{E}} \|\mathbf{B}_{\gamma_0}(\mathbf{x}^s) - T^*(\mathbf{x}^s)\|_{\mathcal{F}}^2$$

Furthermore considering that $\mathcal{H}$ is as proposed in the paper and using Theorem 2 in [4] we have with high probability that:

$$\underset{\mathbf{x}^s \sim \Omega_{\mathcal{S}}}{\mathbb{E}} \|T(\mathbf{x}^s) - T^*(\mathbf{x}^s)\|_{\mathcal{F}}^2$$

$$\leq 2 \sum_{\mathbf{x}^s \in \mathbf{X}_s} \|T(\mathbf{x}^s) - \mathbf{B}_{\gamma_0}(\mathbf{x}^s)\|_{\mathcal{F}}^2 + \mathcal{O}\left(\frac{1}{\sqrt{n_s}}\right) + 2 \underset{\mathbf{x}^s \sim \Omega_{\mathcal{S}}}{\mathbb{E}} \|\mathbf{B}_{\gamma_0}(\mathbf{x}^s) - T^*(\mathbf{x}^s)\|_{\mathcal{F}}^2$$

(Triangle inequality.)

$$\leq 2 \sum_{\mathbf{x}^s \in \mathbf{X}_s} \left(\|T(\mathbf{x}^s) - \mathbf{B}_{\hat{\gamma}}(\mathbf{x}^s)\|_{\mathcal{F}} + \|\mathbf{B}_{\hat{\gamma}}(\mathbf{x}^s) - \mathbf{B}_{\gamma_0}(\mathbf{x}^s)\|_{\mathcal{F}}\right)^2 + \mathcal{O}\left(\frac{1}{\sqrt{n_s}}\right)$$

$$+ 2 \underset{\mathbf{x}^s \sim \Omega_{\mathcal{S}}}{\mathbb{E}} \|\mathbf{B}_{\gamma_0}(\mathbf{x}^s) - T^*(\mathbf{x}^s)\|_{\mathcal{F}}^2$$

$$((a+b)^2 \leq 2a^2 + 2b^2.)$$

$$\leq 4 \sum_{\mathbf{x}^s \in \mathbf{X}_s} \|T(\mathbf{x}^s) - \mathbf{B}_{\hat{\gamma}}(\mathbf{x}^s)\|_{\mathcal{F}}^2 + \mathcal{O}\left(\frac{1}{\sqrt{n_s}}\right)$$

$$+ 4 \sum_{\mathbf{x}^s \in \mathbf{X}_s} \|\mathbf{B}_{\hat{\gamma}}(\mathbf{x}^s) - \mathbf{B}_{\gamma_0}(\mathbf{x}^s)\|_{\mathcal{F}}^2 + 2 \underset{\mathbf{x}^s \sim \Omega_{\mathcal{S}}}{\mathbb{E}} \|\mathbf{B}_{\gamma_0}(\mathbf{x}^s) - T^*(\mathbf{x}^s)\|_{\mathcal{F}}^2. \tag{10}$$

## 6 Complementary Information on Experimental Protocol for the Domain Adaptation Experiments

Algorithm 2 explains the 2-fold circular validation used for tuning the hyper-parameters and inspired from [5, 6]. In this algorithm $M$ is any model able to bring closer the source and the target. For example, with our linear mapping learned from our regularized OT formulation, we have $M(\mathbf{X}_t) = \mathbf{X}_t$ and $M(\mathbf{X}_s) = \mathbf{X}_s \mathbf{L}$.

---

**Algorithm 2:** Circular validation.

    **input** : $(\mathbf{X}_s, \mathbf{y}_s)$ source examples and their labels, $\mathbf{X}_t$ target examples, $\mathcal{A}_{\boldsymbol{\lambda}}$ a learning procedure using hyper-parameters $\boldsymbol{\lambda}$.

    **output** : Average accuracy of $\mathcal{A}_{\boldsymbol{\lambda}}$.

1 **begin**

2      Split $(\mathbf{X}_s, \mathbf{y}_s)$ in two halves $(\mathbf{X}_s^1, \mathbf{y}_s^1)$ and $(\mathbf{X}_s^2, \mathbf{y}_s^2)$.

3      Learn $M^1 = \mathcal{A}_{\boldsymbol{\lambda}}(\mathbf{X}_s^1, \mathbf{y}_s^1, \mathbf{X}_t)$ and set $\mathbf{y}_t^1$ the pseudo-labels of $M^1(\mathbf{X}_t)$ obtained from a 1NN learned on $(M^1(\mathbf{X}_s^1), \mathbf{y}_s^1)$.

4      Set $s^1$ the accuracy of a 1NN learned on $(M^1(\mathbf{X}_t), \mathbf{y}_t^1)$ and evaluated on $(M^1(\mathbf{X}_s^2), \mathbf{y}_s^2)$ .

5      Learn $M^2 = \mathcal{A}_{\boldsymbol{\lambda}}(\mathbf{X}_s^2, \mathbf{y}_s^2, \mathbf{X}_t)$ and set $\mathbf{y}_t^2$ the pseudo-labels of $M^2(\mathbf{X}_t)$ obtained from a 1NN learned on $(M^2(\mathbf{X}_s^2), \mathbf{y}_s^2)$.

6      Set $s^2$ the accuracy of a 1NN learned on $(M^2(\mathbf{X}_t), \mathbf{y}_t^2)$ and evaluated on $(M^2(\mathbf{X}_s^1), \mathbf{y}_s^1)$ .

7      **return** $\frac{s^1 + s^2}{2}$.

---

## 7 Illustrations on the Moons Dataset for the Domain Adaptation Experiments

In Figure 1 we propose some illustrations of the transformation learned by our approach on the Moons dataset.

## 8 Complementary Information on Gradient Adaptation in Image Editing

In the paper, we build from a technique, denoted *Poisson Image Editing*, that operates in the gradient domain of the image. Hence, the gradients of the selection operate as a guidance field for an image reconstruction based on membrane interpolation with appropriate boundary conditions extracted from

Original (Angle: 20)  Linear $\gamma$  Linear $T$

Non linear $\gamma$  Non linear $T$

Source: $+1$
Source: $-1$
Target: $+1$
Target: $-1$

Original (Angle: 50)  Linear $\gamma$  Linear $T$

Non linear $\gamma$  Non linear $T$

Source: $+1$
Source: $-1$
Target: $+1$
Target: $-1$

Figure 1: Illustrations of our approach on the Moons dataset when the rotation is of 20 degrees (first and second rows) and 50 degrees (third and fourth rows). The transformation $T$ follows closely the transport map $\gamma$ and the shapes of the two moons are well preserved. Furthermore learning a linear transformation is better when the angle is 50 degrees. The shrinkage effect is due to the regularization on the transformation which penalizes complex solutions.

the target image. Let $f$ be an unknown scalar function (usually a component of the color space of the image) defined on a given region of the image $\Omega$. Let $f_t$ be the target image defined everywhere apart from the interior of $\Omega$. The Poisson editing method operates by solving for $f$ the following variational optimization problem with Dirichlet boundary conditions:

$$\min_f \int\int_\Omega |\nabla f - \mathbf{v}|^2 \quad \text{with} \quad f|_{\partial\Omega} = f_t|_{\partial\Omega}. \tag{11}$$

Here, $\mathbf{v}$ is the guidance field, which is usually given as the gradient from the source image $f_s$ over the domain $\Omega$, *i.e.* $\mathbf{v} = \nabla f_s|_\Omega$. One can show that the unique solution to this problem is the solution of the following Poisson equation [7]:

$$\Delta f = \text{div } \mathbf{v} \quad \text{over } \Omega, \quad \text{with} \quad f|_{\partial\Omega} = f_t|_{\partial\Omega}, \tag{12}$$

Figure 2: Complementary Illustrations of seamless copies with gradient adaptation.

Figure 3: Illustration of failure of style adaptation.

where div stands for the divergence operator. Using appropriate first order discretization of the Laplacian operator, solving for this problem amounts to solve a big sparse linear system, which can be performed efficiently with multi-grid solvers. We propose in the paper to enhance the generality of this technique by forcing the gradient distribution from the source image to follow the gradient distribution in the target image. We start by learning a transfer function $T_{s \to t} : \mathbb{R}^6 \to \mathbb{R}^6$. We then solve for the following system:

$$\Delta f = \text{div } T_{s \to t}(\mathbf{v}) \quad \text{over } \Omega, \quad \text{with} \quad f|_{\partial \Omega} = f_t|_{\partial \Omega}. \tag{13}$$

In Figure 2 and 3 we show other results produced by our methods. Figure 3 illustrates one particular case of failure of style adaptation: as our method does not modify the spatial arrangement of the gradient, it is not possible to produce the same vast swaths of colors as in the target image.