[Reviews · NeurIPS 2016]

Reviewer 1

Summary

The authors study the optimal transport problem, motivated by the desire to come up with a model which better generalizes to unseen examples. Standard techniques do not generalize as the transport map is not defined for new data and needs to be recomputed whenever new data point is obtained. To this effect, the authors propose an optimization model which enables the joint learning of the transport map and the probabilistic coupling. The (visual) advantages of the new model are validated on domain adaptation and image editing (face swapping) problems. The objective/loss function in new model is, by design, jointly convex in the two variables (transport and coupling), and an alternating minimization method is employed to solve it.

Qualitative Assessment

1) The paper reads well and the new model for joint optimization of transport and coupling looks promising. The computational approach satisfactorily demonstrate the usefulness of the new model and the solution approach. 2) The authors compare to the work of Perez et al [28] which dates back to 2003. Was there no progress in the field in the last 13 years? A detailed commentary on why comparison is done against [28] and why not against newer approaches is needed. 3) It should be explained in words what (1) means, also explaining each term appearing in (1). The current text does not explain the meaning well. It was not explained what notation # means; all used notation should be explained for the paper to be accessible to a wider audience. 4) What is {\cal P} in Line 75? This was not explained – notation is used without any word of warning or advice. 5) Equation (4) should be explained in more detail to a lay audience. What is being mapped to what? Are we mapping index i to x\in \Omega? What is \Omega (this was never defined)? 6) Line 93: Why do you focus on uniform samplings? What do you mean by uniform samplings? This should be explained at more length as it seems to be a critical assumption/limitation of focus. 7) The choice and meaning of the various constants appearing in the new model (6) is not discussed. For instance, why should one normalize by max(C)? After all, lambda_\gamma can account for this. Why should one normalize the regularizer by d^2? Again, \lambda_T can absorb this term. How about the factor of 1/d in front of the first term? 8) On lines 116-118 it is claimed that since the loss in (6) is jointly convex, alternating optimization will converge to the optimal solution. First of all, you need to prove that an optimal solution exists (first argue boundedness below by 0, and so on). Second, even if optimal solution exists, alternating optimization (or block coordinate descent with 2 blocks) need not converge. A simple example of this is min max{|x_1|, |x_2|}. Starting from x=(x_1,x_2) = (1,1), no coordinate direction leads to descent. Hence, the method can get stuck at a non-optimal point. This illustrates that additional conditions to convexity are needed for block coordinate descent to work. Are these satisfied for this function? The paper would be much stronger if this was established. 9) I do not understand the assumption on H in Theorem 1. The assumption seems to be a definition of convex combination of 2 functions. Hence, it is not an assumption! What should be assumed is that H is itself convex. 10) The only theoretical result in the paper, Theorem 1, is a trivial observation. The paper contains a number of typos and English usage issues; some of which are listed below. The authors should re-read the paper and do corrections. In general though, the paper is well written. 1) Line 29: bound -> bounds 2) Line 31: Most of -> Most 3) Line 36: from -> for 4) Line 37: have -> find 5) Line 47: to an -> to 6) Line 51: easily kernelized that -> easily kernelized, which 7) Line 53: evidence on -> evidence for 8) Line 54: interest -> usefulness 9) Line 54: on domain -> in domain 10) Line 56: remaining -> rest 11) Line 71: assuming for -> assuming 12) Line 73: of the -> of 13) Eq (2) should end with a full stop 14) Line 77: writing -> formulation 15) Eq (3) should end with a comma 16) Line 86: computed ->computed, 17) Eq (4) should end with a comma 18) Eq (4): Presumably, \Omega should be replaced by \Omega_{\cal S} 19) Line 93: an -> a 20) Line 93: hence -> , whence 21) Eq (6): Defined max(C) – is this the maximal element in matrix C? This was not defined 22) Eq (6) should end with a comma 23) Line 111: the -> The 24) Line 114: a block -> block 25) Line 114: [16] where -> [16], where 26) Line 122: block coordinates approach -> block coordinate descent approach 27) Line 125: problems -> problem I stopped checking for typos beyond this – but I expect the same frequency in the rest of the paper. Please correct them all. I just did a very cursory check beyond this in the Supplementary: 1) Line 3: Theorem -> theorem 2) Line 9: closeness -> proximity 3) Second inequality after line 14 should be an equality Bibliography: [5, 6, 14, 21, 27]: wasserstein -> Wasserstein [7]: bregman -> Bregman [16]: capitalize journal name [19]: frank-wolfe -> Frank-Wolfe Similar issues appear in biblio for the supplementary part. ---- post-rebuttal remarks ----- I have read the other reviews and the rebuttal and as a result keep my scores unchanged.

Confidence in this Review

2-Confident (read it all; understood it all reasonably well)


Reviewer 2

Summary

This paper attempts to unify two views of the optimal transport (OT) problem. On the one hand, the Kantorovich formulation attempts to find a probabilistic "measure coupling" matching two probability distributions --- relaxation to a probability distribution allows for non-bijectivity, e.g. sending two delta functions in the source to one delta function in the target. On the other hand, the Monge formulation attempts to find an explicit map from points on the source to points on the target; this formulation is preferable for some applications but not always solvable theoretically (e.g. if mass has to split). This paper provides a computational approach halfway between these two views. The basic idea is to augment the Kantorovich OT formulation with a second variable representing the map from source to target. To avoid regularity/existence issues associated with the Monge formulation of transport, the technique restricts the class of maps to a hypothesis class H. A regularization R(.) further prevents crazy results for the new mapping variable. The paper proves that under a (fairly restrictive) affine assumption, the joint optimization problem for the measure coupling and map is convex; coordinate descent algorithms are provided for the linear and kernelized linear cases. Some rough bounds are provided to understand behavior under stochastic sampling, and simple experiments are provided for domain adaptation and an image processing task (the former is certanly ML-related and behaves well; the latter is interesting but probably better suited for the imaging community).

Qualitative Assessment

Even though the idea they present is quite simple, I really like this paper. Our intuition from the smooth optimal transport problem on Euclidean space is that the measure coupling from Kantorovich's formulation encodes a map, but somehow discretization gets in the way of extracting the Monge solution --- essentially the discrete measure coupling (just a doubly stochastic matrix) is too sharp for its own good, clumping points around observed data. The proposed technique gets around this in a simple, usable way. Smaller comments: - There are many English language mistakes. While I didn't find any that got in the way of my understanding, I might suggest finding an editor for the final draft. [I can provide a list of typos for the final draft if needed, but currently am overloaded with reviews and am unable to find time to do this.] - The formulation (6) jointly optimizes for the measure coupling gamma and the map T. This means that there is feedback in both directions --- the choice of gamma affects T and vice versa. My initial guess for how to construct such a map estimation algorithm would be to do this in two steps: compute the *optimal* matrix gamma, then estimate T from gamma. Can the authors provide examples/explanation why this simpler pipeline wouldn't work? Why is it useful to have the regularization R(.) and/or the hypothesis class H affect the optimal transport map T? - Somewhere the background should discuss the Monge-Ampere PDE, satisfied in the smooth case by maps T from optimal transport with quadratic ground distance (under certain regularity assumptions). Also, cite work by A. Oberman on solving this PDE, as it is definitely related to the task at hand -- essentially the Monge-Ampere solution provides a transport map T. - I didn't understand the notation in theorem 1 for the affine property (tT1+(1-t)T2)(x)=tT1(x)+(1-t)T2(x). It seems like this would hold tautologically. Please add a sentence or equation explaining what this means. - L needs to be treated more carefully in (7) and (8). Should it be H = {T : exists L s.t. for all x, T(x)=x^T L} ? Or, is L fixed somewhere? - line 139 ("not moved too far away") -- Isn't this what optimal transport does anyway? - Does the kernelized model in "non-linear transformation" (line 147) satisfy the affine property in theorem 1? - line 169, "three key quantities" -- but, there are four terms in eq(13)? Please clarify, or label the terms in (13) using \underbrace. This analysis is quite nice, however. - line 187 -- This rotated dataset seems tricky for optimal transportation, which doesn't really seek rotations (it seeks potentially nonrigid transformations for which no point moves too far). - The domain adaptation task in sec 4.1 needs additional description. Please describe EXACTLY what problem you are solving as upfront and clearly as possible without assuming the reader is familiar with previous work on OT + domain adaptation. In as simple terms as possible, what is the input and what is the output? What optimization problem do you solve? What is the expected behavior? Some plot figures e.g. illustrating your results (the matching gamma and map T) on a simple dataset like "Moons" would be quite useful.

Confidence in this Review

3-Expert (read the paper in detail, know the area, quite certain of my opinion)


Reviewer 3

Summary

The paper proposes a method for learning the coupling along with an approximation to the transport map. An explicit transformation that approximates the transport is learnt, while the method further addresses the out-of-sample problem. Results in related experiments seem to outperform all compared methods.

Qualitative Assessment

The paper is well-written and results seem promising. Further from experimental improvements, the paper introduces new insights in terms of utilizing barycentric mappings, along with a novel method that accommodates out-of-sample data. I would suggest to the authors that they add more clarity in the description of the method, as it is quite dense and at-times difficult to read (despite the paper being in general well-written).

Confidence in this Review

2-Confident (read it all; understood it all reasonably well)


Reviewer 4

Summary

This paper proposes a new algorithm to jointly learn the coupling and an approximation of the transport map.

Qualitative Assessment

I am not familiar with optimal transport literature. It seems this paper is trying to solve an important problem. Overall the derivation of the algorithm is straightforward and experiment results are promising. It would be better if this paper can provide some theoretical analysis for the solution of the proposed algorithm.

Confidence in this Review

1-Less confident (might not have understood significant parts)


Reviewer 5

Summary

This paper provides a framework for learning the transport map T in an Optimal Transport problem, linked to the Monge problem. Most existing approaches solve this problem by learning a probabilistic coupling \lambda (in the Kantorovich relaxation), then use \lambda to transform examples in the source space to the target space through barycentric mapping. The problem is that the barycentric mapping cannot transform "unseen" examples that are not used during the learning of \lambda. Thus, the paper proposes to jointly learn both \lambda and T (that "approximates" the transport map given \lambda). The paper shows that the proposed framework is jointly convex, provides a bound on the quality of the learned T, and empirically applies the framework to the tasks of domain adaptation and and image editing.

Qualitative Assessment

I am not an expert in Optimal Transport; thus, it is difficult for me to judge technical contributions, the novelty/originality, and potential impact of this work. Also, I have not checked the correctness of the proofs in the supplementary material related to Theorem 1 and the inequality (13). However, if those are assumed correct, there are several strengths: - The paper is well-written. - The paper is well-motivated. In particular, this framework can be applied to many applications that require transformation of "out-of-samples" data points in the source space, or simply used to improve the quality of the transport map. - The proposed optimization formulation seems appropriate --- addressing the problem by learning T directly (though approximately) while still remaining connected to the relaxation. Solutions to both steps in Algorithm 1 seem natural. - Experiments validate the claims. One shows the improved quality of the transport map while the other demonstrates an application that is made possible by the availability of "out-of-samples" transformation. My main concerns are - the usefulness of the inequality (13): Given that it is difficult to bound the second and third quantities, the inequality doesn't tell much about the quality of the learned T. - Eq.(5) and uniform sampling: Can the authors comment on what will happen if the cost function is not the squared l2 distance and if we consider non-uniform sampling. Others: Line 44: One -> On Line 62: \infinity[ to \infinity)

Confidence in this Review

2-Confident (read it all; understood it all reasonably well)